# Optimization of Graphical Parameter Extraction Algorithm for Chip-Level CMP Prediction Model Based on Effective Planarization Length

**DOI:** 10.3390/mi15040549

**Published:** 2024-04-19

**Authors:** Bowen Ren, Lan Chen, Rong Chen, Yan Sun, Yali Wang

**Affiliations:** 1The EDA Center, Institute of Microelectronics of Chinese Academy of Sciences, Beijing 100029, China; renbowen@ime.ac.cn (B.R.); chenrong@ime.ac.cn (R.C.); sunyan1@ime.ac.cn (Y.S.); wangyali@ime.ac.cn (Y.W.); 2The School of Electronic, Electrical and Communication Engineering, University of Chinese Academy of Sciences, Beijing 100049, China

**Keywords:** die-scale CMP model, effective planarization length (EPL), layout-dependent effects, density correction, HKMG, layout extraction

## Abstract

As a planarization technique, chemical mechanical polishing (CMP) continues to suffer from pattern effects that result in large variations in material thickness, which can influence circuit performance and yield. Therefore, tools for predicting post-CMP chip morphology based on the layout-dependent effect (LDE) have become increasingly critical and widely utilized for design verification and manufacturing development. In order to characterize the impact of patterns on polishing, such models often require the extraction of graphic parameters. However, existing extraction algorithms provide a limited description of the interaction effect between layout patterns. To address this problem, we calculate the average density as a density correction and innovatively use a one-dimensional line contact deformation profile as a weighting function. To verify our hypothesis, the density correction method is applied to a density step-height-based high-K metal gate-CMP prediction model. The surface prediction results before and after optimization are compared with the silicon data. The results show a reduction in mean squared error (MSE) of 40.1% and 35.2% in oxide and Al height predictions, respectively, compared with the preoptimization results, confirming that the optimization method can improve the prediction accuracy of the model.

## 1. Introduction

Chemical mechanical polishing (CMP), the preferred surface planarization technique for advanced semiconductor processing, is widely used in the fabrication of high-K metal gates (HKMGs), copper interconnect structures, and dielectric layers. However, during the manufacturing process, the chip surface after CMP exhibits morphological fluctuations. The topographic variation of the chip after CMP is closely related to the layout pattern; this is called the layout-dependent effect (LDE) in the CMP process.

As the process node advances, surface variations after CMP pose numerous challenges in semiconductor manufacturing [1], such as depth-of-focus (DOF) control [2] and device defects [3]. In addition to technology optimization, another approach to improve chip flatness is to simulate and optimize the layout design during the design flow. Through simulations, designers can determine the influence of LDE on manufacturing and optimize the chip design, which is called design for manufacturing (DFM). Then, a DFM tool for CMP simulation based on the LDE becomes essential to ensuring process yield, improving the accuracy of electrical performance prediction and developing layout design rules.

The Massachusetts Institute of Technology (MIT) was the first to incorporate LDE into the CMP topography prediction model. In [4,5,6,7], it was argued that the layout pattern influences the pressure distribution on the chip surface during the CMP process, resulting in a nonuniform material removal rate (MRR) and influencing the final surface topography. Based on this theory, a density step height (DSH) topography prediction model is proposed to determine the influence of the layout pattern density and step height on the pressure distribution. References [8,9] added the influence of the line width and line space information on the CMP topography based on the DSH model. In [10], the linear relationship between step height and MRR was modified to an exponential relationship to improve the accuracy of topography prediction. In addition, in [11,12], the EDA Center of the Institute of Microelectronics, Chinese Academy of Sciences, combined chemical corrosion and mechanical wear mechanisms for an improved topography prediction after CMP. In [13], Bao attempted to predict topography using a convolutional neural network. Although there are differences in the mechanism, the chip-level CMP topography prediction based on the LDE can be divided into the following three processes:Layout grid area division;Grid pattern parameter extraction;Surface profile prediction with pattern parameters.

As mentioned earlier, the prediction of chip morphology after CMP needs to consider the LDE. However, for mathematical models, abstract set structures cannot be directly incorporated into model predictions. The model requires graphical parameters that can represent the features of the patterns as input quantities that affect model predictions, such as the width, perimeter, and density of the patterns. This necessitates the division of the layout into regions, also known as grid area division. The graphical parameters within each grid area are statistically analyzed to serve as the graphical input for the model. However, after dividing the layout into multiple grids, we found that the topography of a single grid was influenced not only by the grid pattern but also by the pattern in adjacent grids. Based on Figure 1, two grids with identical geometric parameters exhibit different surface morphologies after grinding due to the different surrounding geometric environments. Therefore, the pattern parameters extracted from a single grid cannot fully represent the LDE of the grid topography.

To enhance the study of LDE and CMP prediction, we conducted the following work to elucidate the mutual influence among grids.

Proposed the concept of effective planarization length (EPL) based on [14];Hypothesized and analyzed the principles behind the emergence of EPL;Determined the mutual influence weights between grids within EPL.

EPL is defined as the maximum distance between two grids that have a mutual effect during the CMP topography prediction. The longer the distance between two grids, the weaker their interaction. When the distance between the grids exceeds the EPL, there is no interaction between the grid topographies. To reflect this influence in the CMP model, convolution was used to correct the grid pattern density. This approach is similar to that presented in [15]. However, the weighting function presented in [15] does not fit the CMP prediction of an advanced process. In this study, we investigate the contact deformation between the pad asperity and chip surface to obtain a reasonable weight function. Finally, we innovatively use the one-dimensional line contact deformation function as the grid weight function, which is introduced in Section 3. The density after the correction is called the effective density. Compared with the pattern density, the effective density can better reflect the influence of the LDE on the morphology. We apply this density correction algorithm to an existing HKMG-CMP topography prediction tool based on the DSH model, as indicated in Figure 2. The shape prediction data of the tool are compared with the atomic force microscopy (AFM)–measured data before and after density correction, which confirms that density correction can improve the prediction accuracy of the model.

The structure of this paper is as follows: Section 2 introduces the existing DSH chip-level CMP topography prediction models and the problems of existing graphical parameter extraction algorithms. Section 3 defines the EPL and optimizes the model using the EPL. Section 4 introduces the HKMG layout structure and chip surface measurement method. Section 5 compares the model prediction results before and after graphical density correction with the measured data. Section 6 summarizes and discusses the study.

## 2. DSH Model Chip-Level CMP Surface Topography Prediction

The following sections introduce the DSH model and its mechanism, layout extraction algorithm, and MRR calculation. In addition, we discuss the problem found in the layout extraction and relationship between the problem and the EPL.

### 2.1. DSH Prediction Model

The DSH model was first proposed by MIT in [6]. It calculates the remove rate (RR) with a pressure distribution, as indicated in Equations (1) and (2).
(1)$$P_{up}=\left\{\begin{array}{cc} P_{0}+P_{0}\left(\frac{1-\rho }{\rho }\right)\frac{H}{D_{max}}, & 0\leq H\leq D_{max},\\ \frac{P_{0}}{\rho }, & H\geq D_{max} \end{array}\right.$$
(2)$$P_{down}=\left\{\begin{array}{cc} P_{0}(1-\frac{H}{D_{max}}), & 0\leq H\leq D_{max},\\ 0, & H\geq D_{max} \end{array}\right.$$

$P_{up}$ and $P_{down}$ denote the pressure carried in the higher and lower regions of the grid, respectively. *H* is the step height and represents the average difference between the higher and lower regions. $P_{0}$ is the average load of the abrasive pad. ρ is the proportion of the graphical area of the higher region, also known as graphical density. $D_{max}$ is the fitting parameter related to the line width (w) and space (s). In Figure 3, w, s, and H are demonstrated visually. After obtaining the pressure distribution, the MRR is calculated using the Preston formula [16], as indicated in Equation (Equation 3). In Equation (Equation 3), K, α, and β represent different fitting factors. P represents the pressure calculated using Equations (1) and (2), and V represents the relative speed between the pad and wafer in the CMP process.
(3)$$MRR=KP^{\alpha }V^{\beta },$$

### 2.2. Layout Pattern Parameter Extraction

In the actual layout design, there is virtually no standard trench structure with the width and space required by the DSH model. Therefore, it is necessary to equate the actual complex grid pattern with graphic parameters such as line width, spacing, and density. We use the method described in [13]. For each grid, the pattern parameter extraction process is summarized in the following four steps:Count the perimeter $C_{i}$ and area $S_{i}$ of each polygon in the grid figure, where *i* is the number of the polygon in the grid.Solve Equation (Equation 4) to obtain the length $Q_{il}$ and width $Q_{iw}$ of the equivalent rectangle for each polygon in turn by rectangular equivalence:
(4)$$x^{2}-\left(\frac{C_{i}}{2}\right)x+S_{i}=0.$$Calculate the average line widths of all polygons in the grid using the polygon areas as weights:
(5)$$W=\frac{\sum_{i=1}^{n}Q_{iw}S_{i}}{\sum_{i=1}^{n}S_{i}}.$$Calculate the density of graphs in the grid ρ; *D* is the length of the grid side.
(6)$$\rho =\frac{\sum_{i=1}^{n}S_{i}}{D^{2}}$$

The statistical parameters of the graphical parameters are used to calculate the DSH model for computing the mesh morphology.

### 2.3. EPL and Density Correction

During the tests, we identified several problems with the pattern parameter extraction and topography prediction. Intuitively, a smaller grid size and a higher resolution lead to a higher accuracy of the parameter extraction, which should improve the accuracy of topographic prediction. However, the experimental results demonstrate the opposite trend. Reducing the grid size does not improve the accuracy of topography prediction. Conversely, a small grid size reduces the prediction accuracy. We believe that the reason for this phenomenon is that the surface topography within a single grid region is influenced by the pattern within the grid and the pattern of the adjacent grid region in CMP. As indicated in Figure 1 shown in Section 1, the AFM data confirm that the grid topography is influenced by the surrounding patterns. When the grid size is reduced, even if the extracted graphical information is more accurate, the result of the topography prediction within the grid becomes inaccurate because the influence of the surrounding grid graphical information is ignored. As indicated in Figure 4, compared with (a), the grid size in (b) is smaller because the grid surface morphology is jointly influenced by the pattern in the surrounding L range. Therefore, in (a), the parameters extracted from (b) ignore more information regarding the surrounding area pattern, resulting in a lower prediction accuracy for the (b) surface. We refer to L as the EPL, which is the maximum distance between two grids where there is a mutual influence.

To add the influence of the surrounding area pattern to the prediction model, we weighted and summed the grid graphic densities within the EPL after the graphical parameter extraction process introduced in Section 2 and Section 3 to obtain the modified grid graphic density, which is called the effective density of the central grid. We believe that the effective density can better reflect the influence of the LDE on the morphology than the graphic density. In the next section, we present the physical meaning of the EPL and determine a new equation based on its physical meaning as a weighting equation to calculate the effective density.

## 3. EPL and Weighting Function

The following sections introduce the weighting function used for the density correction. Referring to the method introduced in [15], we propose a new weight equation that considers asperity deformation. In addition, we provide a superior explanation of the effective flattening length.

### 3.1. One-Dimensional Line Contact and Partial Deformation of Pad Asperity

During the CMP process, the pressure distribution on the chip surface due to the deformation of the pad is an important factor influencing the chip topography. The deformation generated by the contact between the pad and grid surface structure propagates along the pad to the periphery, which in turn influences the contact between the grid and the pad at other locations of the chip. Therefore, we believe that the EPL and weighting functions are related to the pad deformation profile.

Compared with [15], we believe that the grid-to-grid interaction is more relevant to partial deformation. From the topographic data, the interaction distance between the grids is considerably smaller than the wafer size and closer to the asperity size observed in [17]. Therefore, it is unreasonable to use the entire polishing pad deformation instead of asperity deformation as the weight function. However, calculating the deformation of pad asperities is reasonably complex and involves multiple factors, such as the topography of the contact surface, graphical dimensions, pressure load, Young’s modulus of the pad, Poisson’s ratio, and the size of the asperities. To describe this deformation, [18] assumed that the deformation of asperity satisfies the Hertz contact, and [19] considered the effect caused by a random asperity size. Both are valuable, yet overly complex to optimize a chip-scale CMP prediction tool. To simplify the model, we investigate a single asperity deformation in contact with the trench structures as an approximation of the actual deformation profile. The asperity deformation in contact with the trench is displayed in Figure 5a. The single feature in contact with the asperity is a slender rectangular structure along the y-direction, with the width of the rectangle being considerably smaller than the asperity size. At this contact, the rectangular structure applies a linear load P to the asperity. This contact satisfies the following conditions:The size of the structure in the x-axis direction is considerably smaller than the size of the rough peak.There is no change in the distribution of the structure in the y-axis direction.

Therefore, this contact satisfies the conditions for the application of the one-dimensional linear contact deformation model presented in [9]. The deformation distribution of the plane in the normal direction along the x-axis satisfies Equation (Equation 7). The asperity deformation is illustrated schematically in Figure 5b.
(7)$$U\left(x\right)=-2P\frac{1-\nu^{2}}{\pi E}ln\left(\frac{R_{0}}{x}\right)$$

U(x) is the deformation variable of asperity, *P* is the pressure per unit length applied to the contact surface, ν is Poisson’s ratio of the abrasive pad material, *E* is Young’s modulus of the abrasive pad, and $R_{0}$ deformation is the absolute value of the coordinate of the x-axis when $R_{0}$ deformation is zero. $R_{0}$ has a similar concept with the EPL.

### 3.2. Weight Function and EPL

The deformation of the asperity caused by contact with a single trench feature satisfies one-dimensional line contact. Conversely, the partial deformation induced by the contact of the pad with the trench structures can be considered a result of the superposition of the one-dimensional line contact deformations induced by the contact of the pad with multiple structures, as indicated in Figure 6. In Section 3.1, we demonstrate that the interaction between the grids during the CMP process is related to the pad’s partial deformation. Therefore, we use the one-dimensional line contact deformation profile as a weighting function to describe the grid interaction. The weighting function is in the form of Equation (Equation 8), where $f_{w}$ is the weighting value, *L* is the EPL, and m is the normalized fitting parameter.
(8)$$f_{w}=mln\left(\frac{L}{x}\right)$$

The weighting function is discretized as the corrected weight of the grid density. Because the value of the weight function is infinite at x=0, the value of the weighting function at x=0.5D is used as the value of the center grid weight after discretization. The weights of the surrounding grid are calculated in the order of x=1D,2D,⋯,nD, where *n* depends on $\frac{L}{D}$. Weighting correction to the density is essentially a convolution process, and the EPL determines the size of the convolution kernel. After determining the weights, the relationship between the weighting function and convolution kernel is displayed in Figure 7. The corrected effective density of the grids can be expressed as Equation (Equation 9). In Section 4, we introduce the layout design for measuring the EPL.
(9)$$\rho_{eff}=\rho ⊗f_{w}$$

## 4. Layout Design and Experimental Measurement Method of EPL

To obtain the EPL and specific weighting function described in Section 3 and verify the effect of the correction algorithm on the CMP prediction result, we designed the layout of the HKMG structure, as displayed in Figure 8. Test chips were fabricated by Semiconductor Manufacturing International Corporation (SMIC) with a subset of the process steps (to the formation of the replacement of metal gates). After the final CMP process, the chip surface profile AFM data were used to obtain the EPL and verify the effect of the density-corrected algorithm on the accuracy of the model prediction. The layout parameters are listed in Table 1.

As indicated in Figure 8, layout areas with different parameters were marked with serial numbers from L1_1 to L2_4. In this study, we selected the transition topography from L1_1 to L1_2 as indicated in Figure 9 to introduce our experiment. From the topography data, it is clear that the topography at the center of the array is stable, which means that the height of the Al gate does not fluctuate in the general trend. However, the topography of the transitional region between the arrays indicates an apparent trend, which is caused by the influence of different patterns on adjacent arrays. Therefore, the range of the transitional topography region reflects the range of interactions between grids, which is equal to the EPL. Figure 10 displays the relationship between the range of the transitional topography region and the EPL.

Table 2 displays the measured length of the transitional topography regions; the average length of these regions was 44.5 µm. In this paper, we implemented the topography prediction model using a 10 µm sized grid for pattern parameter extraction. Considering the relationship between the EPL and the edge length of the grid, the EPL for the grid density correction was set to 25 µm. Using Equation (Equation 8), the weighting convolution kernel for the grid density correction was obtained. After ensuring the EPL and weighting kernel, we added the density correction algorithm to the HKMG CMP topography prediction tool to verify the accuracy of the topography prediction results.

The EPL is an important parameter for the optimization algorithm. Before the measurements, we attempted to calculate the EPL without using silicon data to improve the applicability of the model. Unfortunately, it is difficult to build a model to calculate the EPL using the CMP processing parameters. If DFM optimization for every batch of chips relies on the premeasurement of the EPL, this optimization method would become virtually impossible to apply because of cost. However, according to [15], EPL is only relevant to the CMP process, not to graphics or layout design. This means that, for a particular CMP process, regardless of the layout, the EPL remains the same. Therefore, for other types of geometries and configurations, if the process nodes are the same, the previously mentioned optimization algorithm can be applied.

To verify the effect of the density correction algorithm on the prediction results of the CMP prediction model, the prediction results of the model before and after density correction were compared with the measured AFM data. Figure 11 displays the statistical approach for the surface topography data. The topography statistics use a µm observation window consistent with the model grid size to extract the average height of the Al gate within the observation window. Because the AFM profile data indicate the relative amount of variation in the surface topography, the absolute thickness of the Al grid cannot be measured in the absence of a benchmark. In this study, the following test benchmark was used: the average height of the Al gate in the first observation window on the left side of the transitional topography area was used as the baseline, and the relative height of the Al gate in the other observation windows was counted relative to the baseline. The same statistical method was applied to the relative heights of the oxide layers.

To reduce measurement error, we attempted to study the data noise caused by the AFM equipment. According to the data analysis and research in [20], probe vertical drift and probe tilt were the main causes of noise during the AFM test. The vertical drift of the probe is caused by the mechanical vibration of the probe during CMP, which causes high-frequency noise in the data. To solve the high-frequency noise, we observed the energy frequency distribution of the AFM data from the frequency domain and performed noise reduction on the AFM data by setting the energy threshold. A probe tilt indicates that the probe is not vertical across the chip surface. This type of equipment error causes a low-frequency noise. We calculated the average gradient of AFM data to remove the influence of low-frequency noise caused by probe tilt. In addition to the AFM data processing approach, we are also considering reducing the errors by changing the measurement method. For example, several AFM scans and averages of the resulting profiles can reduce the observation error effectively. Moreover, we are considering other surface measuring methods such as an optical profilometer as a validation of AFM to better observe the surface profile. In Section 5, the measured relative heights of the Al and oxide are compared with the model predictions to confirm that density corrections can have a positive effect on the CMP prediction tool.

## 5. Data Comparison and Result Analysis

The effect of the density correction on the accuracy of the model prediction results was observed by comparing the surface topography data displayed in Section 4 with the predicted data from the chip-level CMP topography prediction model. Figure 12a,b display the predicted values of the oxide and Al gate relative thicknesses in the six regions compared with the measured values. The prediction accuracy of the model before and after the density correction was determined using the root mean square error (RMSE). After the observation made in Section 4, the AFM data of the transectional topography were divided into six windows and the Al and oxide heights for each window were obtained. For Al, for example, the RMSE was calculated using Equation (Equation 10), where $A_{i}$ represents the Al height of each window measured from the AFM data and $B_{i}$ is the simulation value of the optimized CMP model. In Equation (Equation 10), *n* represents the number of observation windows, which depends on the EPL and grid size. The RMSE of each region are listed in Table 3.
(10)$$RMSE=\sqrt{\frac{\sum_{i=1}^{n}{\left(A_{i}-B_{i}\right)}^{2}}{n}}$$

From the comparison of the relative height and RMSE, it is clear that using the density correction algorithm to correct the grid pattern information extraction of the CMP topography prediction model can effectively improve the prediction accuracy of the model. The corrected grid pattern density can better reflect the influence of the LDE on the CMP surface topography in the transitional regions. Overall, after adding the density correction, the RMSE of the model prediction of the relative heights of the Al grid and oxide layer was smaller. This demonstrates that density correction improves the model prediction accuracy.

Table 4 displays the variation in the RMSE for each region. By region, the RMSEs of the oxide and Al layer heights were reduced; however, the percentage of reduction was not the same. For example, the RMSE improvement effect in L2_1–L2_2 was not evident. We believe that this is because of the small differences in the layout parameters designed for the L2_1 and L2_2 regions, for which the width/line was (2.5 µm, 2.5 µm) and (3 µm, 2 µm), respectively. Because of the small differences in the pattern, the surface topography tends to be less undulating after CMP. This leads to random errors, which have a more serious effect on the on-chip surface. As the tendency of the surface undulate decreases, the share of the measurement error in the surface profile measurement increases. This random measurement error cannot be resolved according to the CMP model system optimization method. However, for the L2_1–L2_2 topography data, this can be considered a bad data point, which introduces excessive low-frequency noise into the measurement process such that the measured data are distant from the chip surface. Consequently, a comparison between the predicted and measured results in this region becomes meaningless, and the topography prediction becomes difficult to match to the measured data with noise pollution.

Regarding the measured morphology data, it should be noted that not all areas with “small differences in design parameters” are not significantly optimized. For example, region L2_3–L2_4 has similar design parameters to those of L2_1–L2_2, yet the optimization effect in this area is clear. For this phenomenon, we argue that the effect of low-frequency noise on the observations is randomized. With this random factor interference, the credibility of the surface prediction is low. It is clear that the optimization of the L2_3–L2_4 region is well above average. This overly strong or weak optimization effect, influenced by noise, is not credible. Therefore, in the subsequent discussion, we consider regions L2_1–L2_2 and L2_3–L2_4 as bad data points and exclude them. Moreover, we must solve for the low-frequency noise reduction. However, low-frequency noise does not have as distinct a frequency characteristic as high-frequency noise does. Thus, we could not determine whether the signal components in the AFM data were low-frequency noise or raw morphologies. The cross-validation of multimode measurements could be a solution to this problem. For example, scanning electron microscopy is used to measure the topography of the same area and compare it with AFM data to identify and eliminate low-frequency noise.

From Table 4, we can observe that the modified model optimizes differently for different material thickness predictions. The predictions of the oxide thickness were generally more optimized than the thickness predictions for Al in the same region. We believe that this phenomenon was triggered by the design of the density correction weight function. When designing the weight function, we referred to the following contact deformation equation. Clearly, after CMP, the surface heights of the oxide were higher than those of Al. Therefore, the deformation of the pad asperity caused by the oxide matches the contact deformation equation more closely than the deformation caused by Al. Therefore, after applying this weighting equation for density correction, the optimization of the oxide outperformed that of Al. This also implies that the weighting equation is not perfect, and that the method can be further optimized in terms of the design of the weighting equations.

In addition to the RMSE, we calculated the related coefficient (R) between the profile data and model-predicted value before and after optimization to verify whether the prediction results of the optimized model could better describe the chip morphology trend. The calculation method for the related coefficient (R) is displayed in Equation (Equation 11), where $A_{i}$ and $B_{i}$ represent the surface AFM data and model-predicted values, respectively; Cov represents the covariance; and Var is the variance. Table 5 lists the related coefficient results before and after applying the optimization method. It is clear that, after density correction, the predicted topography is closer to the chip data in terms of variation trend. Even in Region L2_1–L2_2, there is a better-related coefficient. This means that, in L2_1–L2_2, even if the predicted topography indicates unclear optimization, the trend of the predicted value is closer to that of the measured value after density correction.
(11)$$R=\frac{Cov(A_{i},B_{i})}{\sqrt{Var\left(A_{i}\right)Var\left(B_{i}\right)}}$$

In conclusion, the density correction method based on the EPL described in this study has certain limitations in the acquisition of the weight function. However, it can improve the prediction accuracy of the DSH-model-based HKMG CMP surface topography prediction tool. In addition to improving the prediction accuracy of the existing model, this method demonstrates the interaction between grid patterns. The study of grid interactions becomes increasingly important as the grid size decreases with the chip parameter size. Moreover, in the comparison and validation of silicon data, the noise problem had a significant influence on the optimization of the model. The study and solution of the noise problem will become indispensable in the future for the building and optimization of a CMP prediction model.

## 6. Conclusions and Prospect

In this paper, a density correction method for the CMP surface topography prediction model based on the EPL was proposed. This method corrects the grid density in the DSH model by considering more pattern information of the area within a certain range around the grid, referred to as the EPL. Based on the partial deformation of the pad during the CMP process, a one-dimensional line contact deformation profile is used as a weighting function to correct the grid density. This correction method was applied to a DSH-model-based HKMG CMP topography prediction tool, and the predicted data before and after correction were compared with the measured surface data. After excluding bad data due to measurement noise, the results indicated that the prediction results of the density correction model were closer to the actual surface topography data, and the RMSEs of the Al and oxide layer relative thicknesses were reduced by 40.2% and 35.2%, respectively. Moreover, the related coefficient indicated that the predicted data after correction exhibited a trend closer to the AFM data than before. It was demonstrated that the density correction method based on the EPL can improve the prediction accuracy of the CMP topography prediction model.

In subsequent studies, the acquisition of a weighting function is a direction that deserves investigation. The weighting function describes the degree of interaction between the grids within the EPL. Therefore, weight function is an indispensable part of the process of studying the EPL. The choice of the weight function in this study is justified yet not completely reasonable, and the experimental data also reflect defects in the weighting function. In addition, from the experiment, the problem of noise in the measurement data is a non-negligible issue in CMP modeling. The study of noise reduction in raw data during CMP modeling is also an important direction for future research.

As the feature size decreased, the effect of structural interactions during the CMP process became more pronounced. Any modeling algorithm involving layout partitioning inevitably must consider intergrid interactions. Therefore, the concept of EPL, which describes the extent of grid interactions, is not limited to CMP prediction model correction. This concept can also be applied to dummy fill research to consider the influence of the interaction effect on dummy fill. A high-quality dummy fill can reduce the number of defective points caused by graphical effects in the CMP process. This is an indispensable part of layout correction. Therefore, the next step of this work will focus on both the internal weighting function acquisition and its application in the field of dummy filling.

## Figures and Tables

**Figure 1 micromachines-15-00549-f001:**
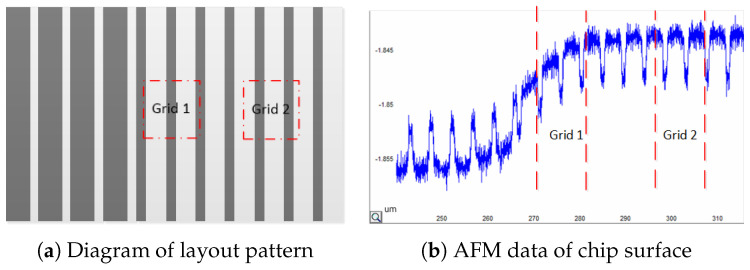
Grids 1 and 2 have the same grid pattern characteristics; however, the surface profiles of Grids 1 and 2 are different owing to the different surrounding pattern environments.

**Figure 2 micromachines-15-00549-f002:**
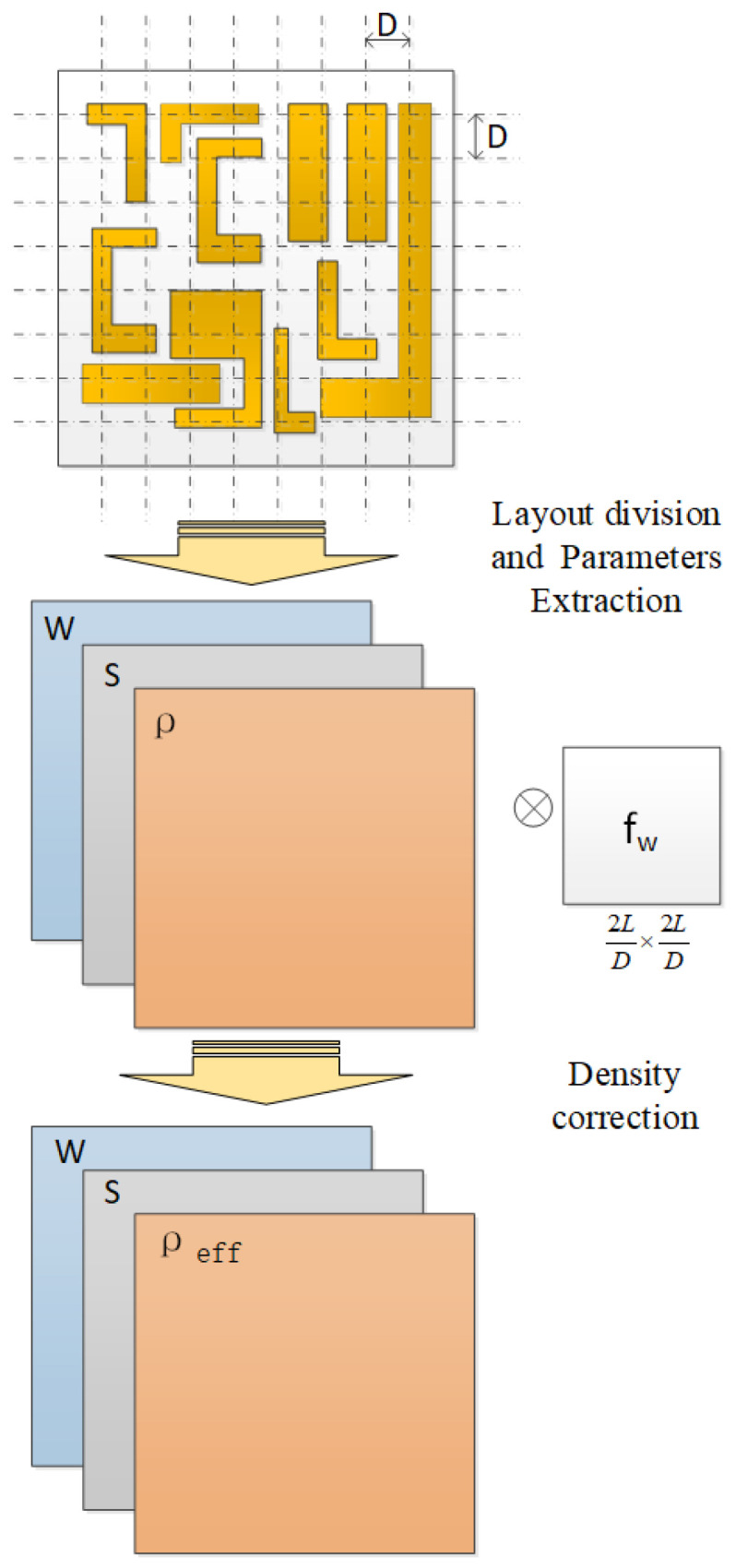
Diagram of density correction flow containing layout division, parameters extraction, and density correction. D is the side length of the grids. L is the EPL. W, S, and P represent the layout parameters as width, space, and density, respectively. $f_{w}$ is the correction matrix. After density correction, the pattern parameters are used in chip surface prediction.

**Figure 3 micromachines-15-00549-f003:**
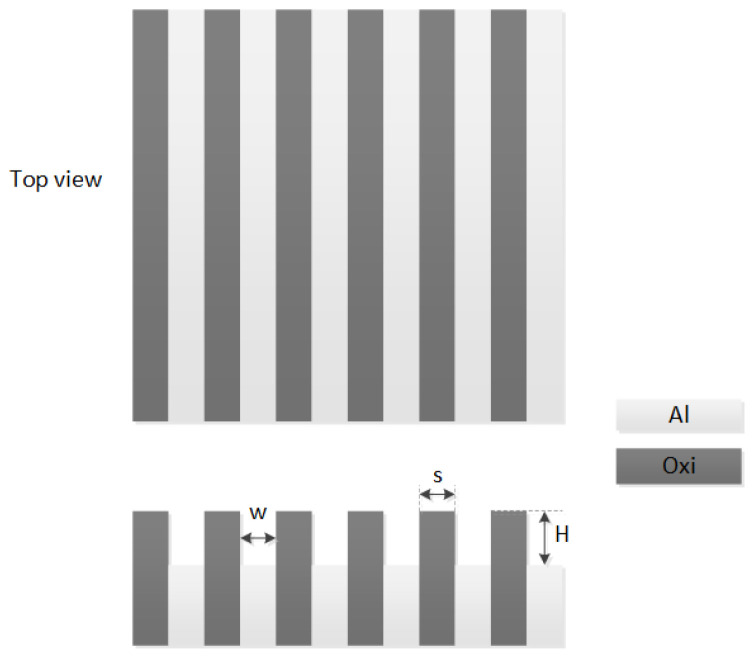
Diagram of line/space structure and definition of pattern parameters.

**Figure 4 micromachines-15-00549-f004:**
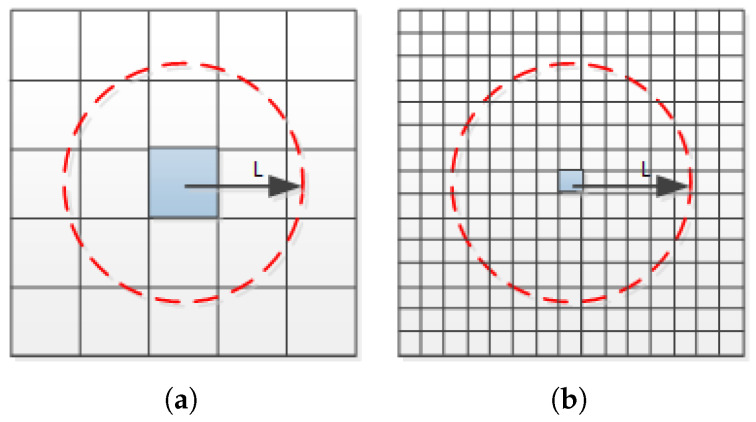
CMP topography of the central blue grid region is influenced by the graphs within the surrounding L. Compared with (**a**), in (**b**), the grid density is extracted more accurately; however, the prediction accuracy is lower, setting L as the EPL.

**Figure 5 micromachines-15-00549-f005:**
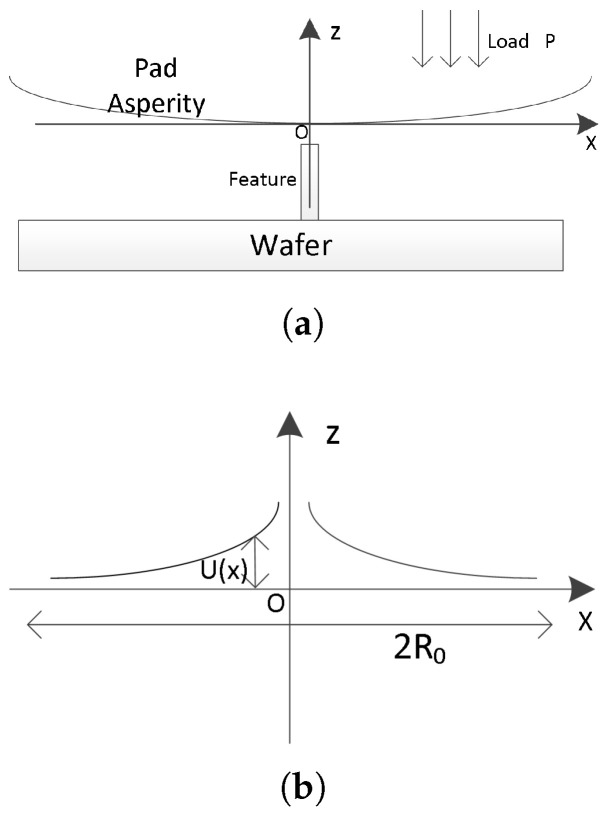
(**a**) Schematic diagram of line/space single graphical structure with pad asperity contact; (**b**) schematic diagram of one-dimensional line contact deformation.

**Figure 6 micromachines-15-00549-f006:**
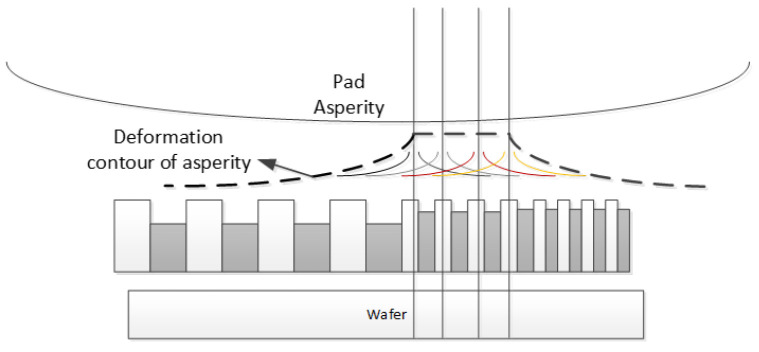
Abrasive pad and complex surface pattern contact deformation contour diagram.

**Figure 7 micromachines-15-00549-f007:**
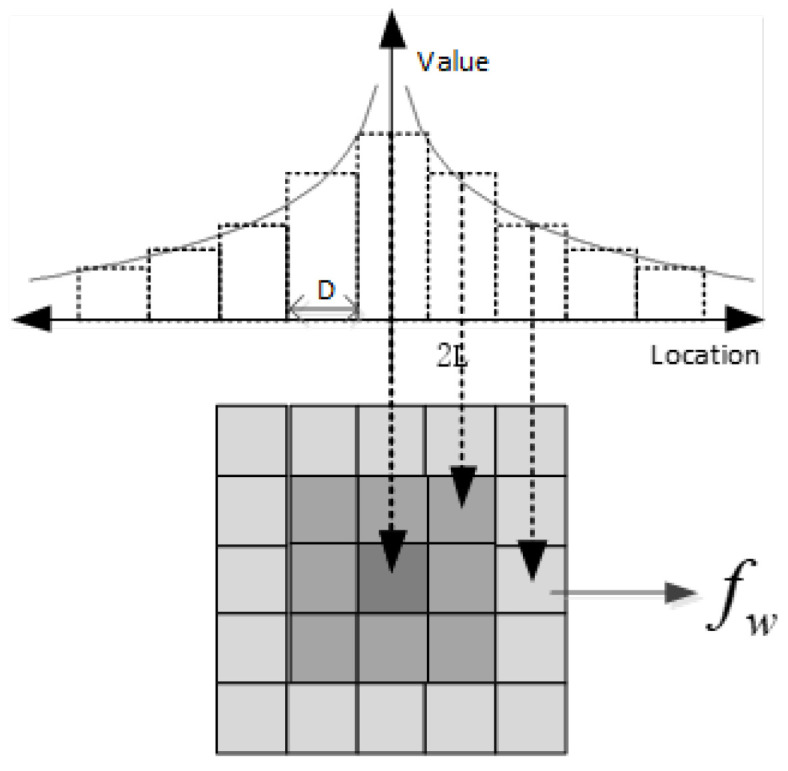
Weight matrix after discretization of weight function.

**Figure 8 micromachines-15-00549-f008:**
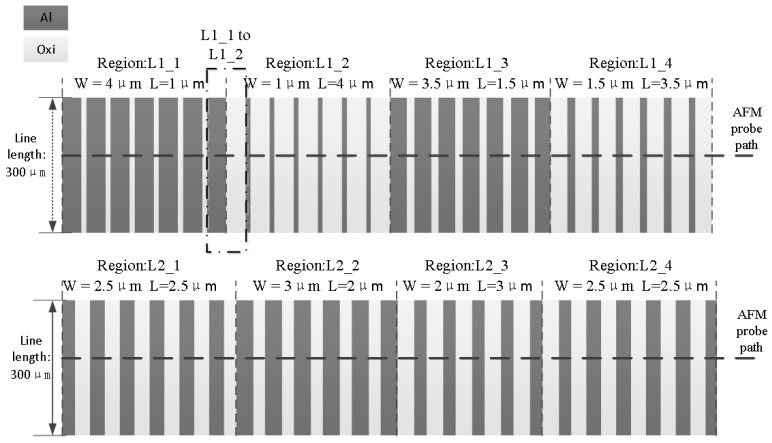
Test layout design and AFM measurement path. The focus is on measuring the surface topography within the transition profile area.

**Figure 9 micromachines-15-00549-f009:**
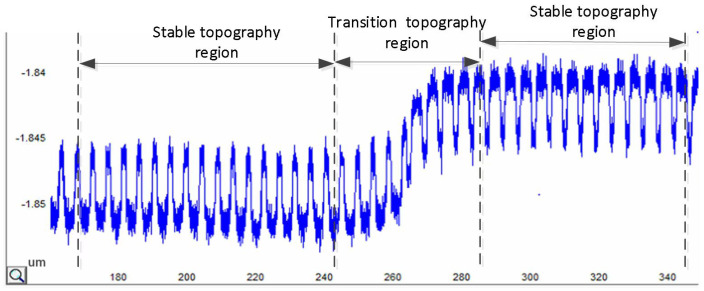
Schematic of AFM surface topography profile.

**Figure 10 micromachines-15-00549-f010:**
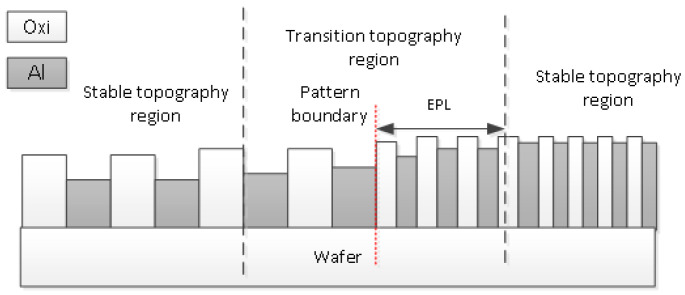
Schematic diagram of the relationship between the transitional topography region and the EPL. The variation in surface topography is caused by the different pattern on adjacent arrays. The distance from the position at the array edge to the stable topography region represents the range of pattern interaction, i.e., the EPL. The EPL is therefore one-half of the transitional topography region range.

**Figure 11 micromachines-15-00549-f011:**
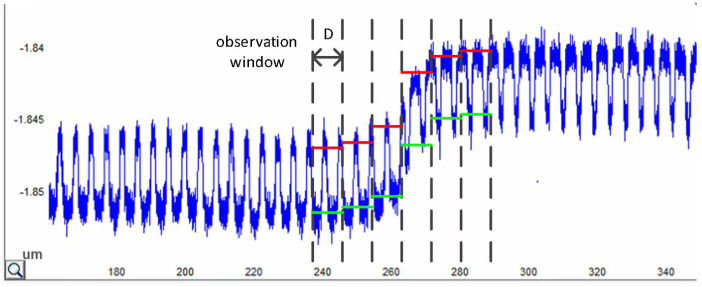
Height measurement method of transection topography area. The dashed line indicates the observation window, which has the same size as the layout grids. The red line represents the observed value of the oxide height. The green line represents the observed value of the Al height.

**Figure 12 micromachines-15-00549-f012:**
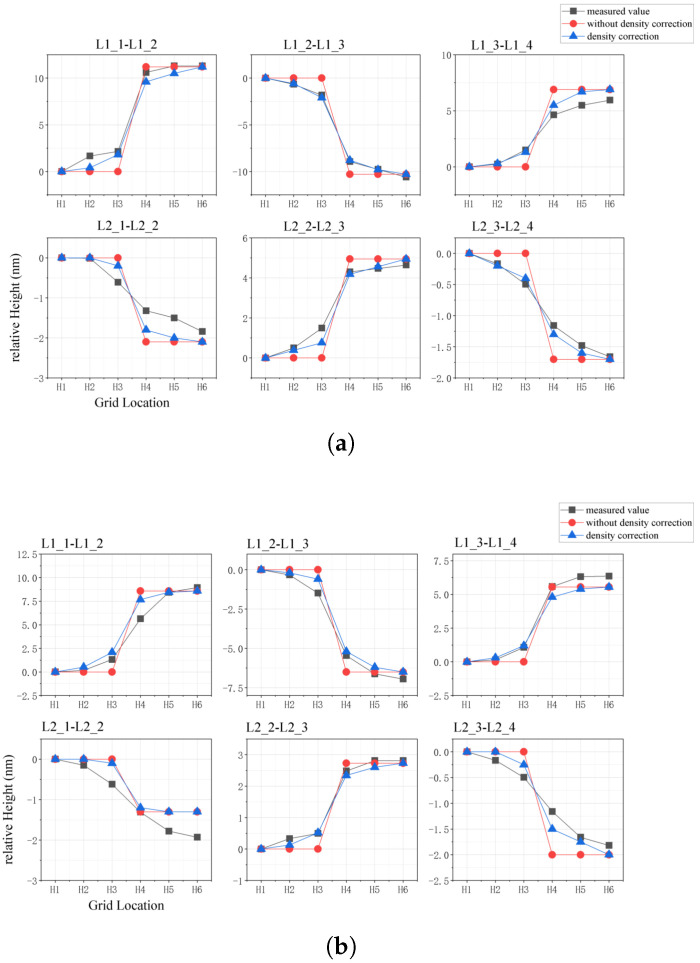
Comparison of relative height data of six transition topography areas: (**a**) oxide height; (**b**) Al height.

**Table 1 micromachines-15-00549-t001:** Layout parameters.

Region Number	Width w (µm)	Space s (µm)	Density ρ (%)
L1_1	4	1	80
L1_2	1	4	20
L1_3	3.5	1.5	70
L1_4	1.5	3.5	30
L2_1	2.5	2.5	50
L2_2	3	2	60
L2_3	2	3	40
L2_4	2.5	2.5	50

**Table 2 micromachines-15-00549-t002:** Measurement of transition area length.

Region	L1_1-2	L1_2-3	L1_3-4	L2_1-2	L2_2-3	L2_3-4
L (µm)	41	43	49	52	42	40

**Table 3 micromachines-15-00549-t003:** RMSE of surface prediction.

Region Number	No Correction Oxide (nm)	Correction Oxide (nm)	No Correction Al (nm)	Correction Al (nm)
L1_1-2	1.137	0.748	1.331	0.912
L1_2-3	0.998	0.179	0.778	0.457
L1_3-4	1.131	0.727	0.682	0.609
L2_1-2	0.484	0.345	0.424	0.394
L2_2-3	0.729	0.331	0.502	0.268
L2_3-4	0.320	0.088	0.434	0.202
Avg	0.811	0.525	0.701	0.474

**Table 4 micromachines-15-00549-t004:** Reduction of RMSE after optimization.

Region Number	Reduction in RMSE	
	Oxide	Al
L1_1-2	34.2%	31.4%
L1_2-3	82.0%	41.2%
L1_3-4	44.6%	22.1%
L2_1-2	28.7%	7%
L2_2-3	54.5%	46.5%
L2_3-4	72.6%	53.4%
Avg	51.5%	35.9%

**Table 5 micromachines-15-00549-t005:** Related coefficient (R) of surface prediction.

Region Number	R of Oxide		R of Al	
	No Correction	Correction	No Correction	Correction
L1_1-2	0.9901	0.9955	0.9551	0.9781
L1_2-3	0.9874	0.9994	0.9761	0.9954
L1_3-4	0.9693	0.9976	0.9841	0.9992
L2_1-2	0.9361	0.9677	0.9486	0.9599
L2_2-3	0.9941	0.9926	0.9888	0.9977
L2_3-4	0.9464	0.9955	0.9806	0.9837

## Data Availability

The data presented in this study are available on request from the corresponding author. The data are not publicly available due to the reason for confidentiality. The original morphology of the chip is confidential and cannot provide detailed morphology data.

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
