# Peer review of "Optimization of Graphical Parameter Extraction Algorithm for Chip-Level CMP Prediction Model Based on Effective Planarization Length"

_micromachines, 2024, doi:10.3390/mi15040549_

Round 1
Reviewer 1 Report
Comments and Suggestions for Authors
The authors calculate the average density as a density correction, and innovatively use a one dimensional line-contact deformation profile as a weighting function. The results shown a reduction in Mean Squared Error(MSE) of 40.1% and 35.2% in oxide and Al height predictions.
The reviewer recommends publishing of the manuscript.
Comments on the Quality of English LanguageMinor editing of English language required
Author Response
Thank you for your efforts in reviewing my article, I will try my best to improve the English of the article!
Reviewer 2 Report
Comments and Suggestions for Authors
The authors present a model for chemical mechanical polishing (CMP) - a critical component of semiconductor processing.
1) Can the authors comment if shear forces on the features during CMP are important for the planarization of features?
2) Can the authors comment if the rotation rates of either the wafer or polishing pad impact the model?
3) Line 66: "Proposeing the concept of effective planarization length (EPL) based on [14]." is a typo which should be spelled as "Proposing"
Comments on the Quality of English LanguageThe quality of English is acceptable.
Author Response
Thank you for your efforts in reviewing my manuscript. Your comments are considered and responded to on a case-by-case basis. You are welcome to discuss any issues of interest with me.
1、Can the authors comment if shear forces on the features during CMP are important for the planarization of features?
A:Shear forces on the features are Important during the CMP process. However, the shear that occurs on features in current processes is primarily fluid shear, i.e., shear generated by the slurry fluid between features. This is due to the fact that in advanced processes, the size of the chip surface features is much smaller than the abrasive pad. In this case, solid-to-solid contact generates forces that are considered to be primarily frictional.
2、Can the authors comment if the rotation rates of either the wafer or polishing pad impact the model?
A: Rotation rates of either the wafer or polishing pad are important for material removal rate of CMP process. The formula for calculating the rate of material removal is presented in Section 2.1 of the article. It can be seen that the material removal rate is related to the relative velocity of the chip to the wafer and the pressure. This relative velocity is determined by a combination of wafer rotation rate, pad rotation rate, and the position of the chip on the wafer.
3、Line 66: "Proposeing the concept of effective planarization length (EPL) based on [14]." is a typo which should be spelled as "Proposing".
A:Thank you for your careful correction, I will modify it in the new version.